# Influence of Nano-SiO_2_, Nano-CaCO_3_ and Nano-Al_2_O_3_ on Rheological Properties of Cement–Fly Ash Paste

**DOI:** 10.3390/ma12162598

**Published:** 2019-08-15

**Authors:** Yiming Peng, Kunlin Ma, Guangcheng Long, Youjun Xie

**Affiliations:** 1School of Civil Engineering, Central South University, Changsha 410075, China; 2National Engineering Laboratory for Construction Technology of High-Speed Railway, Changsha 410075, China

**Keywords:** rheology, cement–fly ash paste, nano-SiO_2_, nano-CaCO_3_, nano-Al_2_O_3_, resting time

## Abstract

Rheological curves of cement–fly ash (C–FA) paste incorporating nanomaterials including nano-SiO_2_ (NS), nano-CaCO_3_ (NC) and nano-Al_2_O_3_ (NA) at different resting times (hydration time of 5 min, 60 min, and 120 min) were tested with a rheometer. The rheological behaviors were described by the Herschel–Bulkley (H–B) model, and the influences of these nanomaterials on rheological properties of C–FA paste were compared. Results show that the types, content of nanomaterials and resting time have great influences on the rheological properties of C–FA paste. Incorporating NS and NA increases yield stress and plastic viscosity, and decreases the rheological index of C–FA paste. When the content of NS and NA were 2 wt%, the rheological index of C–FA paste was less than 1, indicating rheological behavior changes from shear thickening to shear thinning. Meanwhile, with rising resting time, yield stress and plastic viscosity increased significantly, but the rheological index decreased evidently, showing paste takes on shear thinning due to the rise of resting time. However, incorporating 3 wt% NC and the rising of resting time did not change the rheological properties of C–FA paste. These differences are mainly that the specific surface area (SSA) of NS (150 m^2^/g) and NA (120 m^2^/g) are much larger than that of NC (40 m^2^/g). The huge SSA of NS and NA consume lots of free water and these tiny particles accelerate the hydration process during resting time.

## 1. Introduction

As the main by-product generated from coal-firing power stations, fly ash (FA), which mainly consists of active SiO_2_ and Al_2_O_3_, has gradually become an essential component for modern concrete products [1]. The utilization of FA in cementitious material can help to reduce environmental pollution and promote global sustainable development (e.g., reduce the emission of CO_2_) [2,3,4]. Replacing cement by a certain amount of FA can significantly improve workability in the fresh stage and the long-term properties in the hardening stage [5,6,7,8,9]. Fresh cementitious material (FCM) is a complex multiphase dispersion system, which consists of different particles with complex chemical constituents, resulting in different rheological behaviors. The incorporation of FA leads to the decrease of yield stress, plastic viscosity and thixotropy [10], which makes cementitious materials easy pour and cast. As a result of this, the workability of FCM incorporating FA has improved obviously. However, the decrease in plastic viscosity results in instability of FCM, including segregation, bleeding, and coarse aggregate sinking in the fresh stage, which affect the performance of hardened concrete. Therefore, the improvement of stability of fresh cement–fly ash (C–FA) paste is important. 

As an effective means to characterize the nature of workability of FCM, rheology has been applied widely in many engineering materials fields, including self-compacting concrete (SCC), cement asphalt mortar, oil well cement, and 3D printed materials [11,12,13,14]. Rheological properties of FCM are influenced by many factors, such as water-to-binder ratio [15], solid concentration [16], superplasticizer [17,18], shape and fineness of particles [19], et al. 

The application of nanomaterials has developed at an astonishing speed in recent years. The incorporation of nanomaterials has significantly changed cementitious materials properties, not only the workability of the fresh stage but also the development of mechanical properties and microstructures in the hardening process [20,21]. The addition of nano-SiO_2_ (NS) significantly reduces the setting time of cement paste, promotes the formation of more hydration products [22,23,24] and improves the pore structure of concrete [25,26]. Due to its large specific surface area (SSA), the introduction of NS makes the water requirement of cement-based system increase [27,28]. Ouyang et al. [29] proposed an original viscosity prediction model to predict the minimum apparent viscosity of cement paste with different NS dosages and water to cement ratio (W/C). Nano-Al_2_O_3_ (NA) is also widely used as a modified component, and the SSA of NA is close to NS [30]. At the early hydration stage, the initial dissolution of NA leads a higher ettringite content [31]. SEM micrograph reveals the formation of a much denser microstructure with NA addition compared with cement paste [32]. Liu et al. [33] found that nano-CaCO_3_ (NC) had little effect on the water requirement of standard consistency of cement. However, with the increase of NC content from 0 wt% to 2 wt%, the fluidity of cement paste decreased slightly, and the setting time was shortened. 

Studies also have shown that cementitious material is a kind of time-dependent material. The workability and mechanical properties of cementitious materials, including rheological properties change along with resting time [34,35,36]. Aiad et al. [37] found that the addition of chemical admixtures will make a great influence on the rheological properties of cement paste for a relatively long time (120 min). The incorporation of nanomaterials, such as NS and NA, can accelerate the hydration process [38], and then affect the rheological properties of FCM. However, little work has been done on the rheology of FCM containing nanomaterials with the rheological model using a rheometer, and the comparison of the rheology of FCM with different nanomaterials under different resting times is much rarer. In this paper, the rheological properties of C–FA paste containing NS, NC, and NA with diverse content at the different resting times were tested. The Herschel–Bulkley (H–B) model was applied to describe the relationships of shear stress vs. shear rate and apparent viscosity vs. shear rate. The comparison of the rheology of various samples was performed by contrasting their yield stress, plastic viscosity, and rheological index at the different resting time.

## 2. Materials and Methods

### 2.1. Materials and Admixture Proportion

The cement (C) was Ordinary Portland cement with a grade of 42.5 satisfied Chinese Standard GB 175. Qualified and densified fly ash (FA) of Class F had a specific surface area of 463 m^2^/kg and a density of 2.38 g/cm^3^. The composition and physical properties of C and FA are given in Table 1. Particle size distribution of C and FA is exhibited in Figure 1. Physical properties of NS, NC, and NA specified by Hangzhou Wanjing New Materials Co. Ltd. (Hangzhou, China) is shown in Table 2. Polycarboxylic acid superplasticizer (SP) with water-reducing rate 32% and solid content 33.1% was used to enhance the flowability of fresh mixture.

The mix proportions in this paper are given in Table 3. Nine groups of mixtures were prepared, including one C–FA paste incorporating 25 wt% FA. Due to the high viscosity of 3 wt% NS, which was over the range of rheometer, two groups of mixtures were replaced with 1 wt% NS and 2 wt% NS respectively. Moreover, the other six groups of paste were modified with NC and NA, respectively, in different mass increments from 1 wt% to 3 wt%. Each mixture kept the water to binder ratio (W/B) 0.32 and superplasticizer (SP) 0.4%.

### 2.2. Rheology Test

A room with ambient temperature (25 ± 2) °C and relative humidity (70% ± 5%) was used to carry out this test. The MCR 102 rheometer produced by Anton Paar Company in Graz, Austria (seen in Figure 2) was used to determine the rheological curves of different pastes. The rheometer consists of coaxial cylinders with inner and outer radii of 11 and 21 mm, respectively, and the height is 40 mm (see Figure 3). In order to ensure the nanoparticles dispersed uniformly in the C–FA paste, the ultrasonic nanomaterials disperser was used to disperse the mixtures containing water, SP and nanomaterials. After ultrasonic dispersion, cement and fly ash were added and the electric mixer was used to stir the paste (seen in Figure 4). The process of stirring the paste was first stirred for 90 s at a low rate, then stopped for 10 s, and finally quickly stirred for 90 s at a high rate. Before the rheology test, the prepared fresh composite paste was kept in resting time for 5 min, 60 min, and 120 min respectively. The shear rate was then increased gradually from 1 s^−1^ to 300 s^−1^. Moreover, the time interval between each point was controlled to a constant value of 5 s. Rheological equation and related rheological parameters were calculated according to mathematic fitting.

### 2.3. Rheological Parameters Analysis

The rheological properties of cementitious materials are usually simplified by a specific rheological model because of their complexity. Here the Herschel–Bulkley (H–B) model was used to describe the rheology of fresh paste, seen in Equation (1):γ = 0, *τ* < *τ_0_**τ* = *τ_0_* + *K*γ*^n^*, *τ* ≥ *τ_0_*(1)
where *τ* (Pa) is the shear stress, *τ_0_* (Pa) is the yield stress, *K* (Pa·s^n^) is the plastic viscosity coefficient, *γ* (s^−1^) is the shear rate, *n* is the rheological index. It is generally believed that shear thickening occurs when *n* > 1 and shear thinning occurs when *n* < 1. The larger the rheological index (*n*) is, the stronger the intensity of shear thickening is. 

Figure 5 shows two different rheological curves obtained from this test. Figure 5 shows a great difference after the incorporation of NS. With the increasing shear rate, shear stress decreased first (Part A) and then increased (Part B) for C–FA paste incorporating 2% NS. Therefore, in this paper, the yield stress was calculated according to Part A, and the plastic viscosity was calculated according to Part B.

## 3. Results

### 3.1. Influence of NS on C–FA Paste Rheology 

#### 3.1.1. Rheological Curve 

Figure 6a–c presents the rheological curves of C–FA paste with NS at different resting times. As can be seen from Figure 6, along with the increase of shear rate, the shear stress was on the rise, and the rheological curves of C–FA paste incorporating 1% and 2% NS were different from C–FA paste, which indicates incorporating NS into C–FA paste changed the rheological properties of C–FA paste obviously. To begin with, Figure 6a shows that when NS content was 2%, the shear stress enlarged greatly. Second, Figure 6b,c shows that when NS content was only 1%, the shear stress of paste incorporating NS prominently increased with increasing resting time. However, when NS content was 2%, shear stress decreased first and then increased along with the development of shear rate, and the minimum shear rate was 13.8 s^−1^, 21.7 s^−1^, 41.1 s^−1^ at 5 min, 60 min, and 120 min resting time respectively.

#### 3.1.2. Yield Stress

Yield stress (*τ_0_*) is the minimum shear stress that drives materials to initiate flow and deformation. *τ_0_* is produced by adhesive force and frictional force among particles. The value of yield stress is significantly related to the workability of cementitious materials. An increasing yield stress may cause difficulty in self-leveling and vibrating for paste. *τ_0_* of C–FA paste containing NS at different resting times are diagrammed in Figure 7, and Table 4 also shows the rheological parameters fitted from H–B model.

As can be seen from Figure 7, the addition of NS into C–FA paste increased *τ_0_* greatly, and in the same NS content, the longer the resting time, the larger the *τ_0_* was, especially when NS content was 2%. Figure 8 illustrates the influence of resting time on *τ_0_*. Figure 8 shows that *τ_0_* increased obviously with the development of resting time after the incorporation of NS and resting time. The addition of 2% NS also enlarged *τ_0_* at different resting times. Data in Table 4 show that with increasing resting time the *τ_0_* of C–FA paste decreased, but *τ_0_* of paste incorporating NS increased greatly. *τ_0_* of C–FA paste increased from 1.32 Pa to 170 Pa at 5 min when NS content increased from 0% to 2%.

#### 3.1.3. Plastic Viscosity 

Plastic viscosity (*K*), also called flow resistance, indicates the number of micro-structures resisting to flow within materials. It is influenced by the filler shape, filler size, and filler concentration. Moreover, plastic viscosity can influence the flow velocity of fluid, including cementitious materials. Apparent viscosity (*η_0_*) reflects the dynamic change of plastic viscosity under different shear rate.

Apparent viscosity (*η_0_*) of C–FA paste containing NS at the different resting times is described in Figure 9. Results in Figure 9 illustrate that, first, at the same shear rate, the addition of NS into C–FA paste enlarged *η_0_*, especially when NS content was 2%. Secondly, *η_0_* of C–FA paste incorporating different NS exhibited obvious different changes. *η_0_* of C–FA paste with 0% and 1% NS almost increased with shear rate, but *η_0_* of C–FA paste with 2% decreased with the rising shear rate.

Figure 10 shows the influence of NS content on plastic viscosity (*K*). Figure 11 shows the influence of resting time on *K*. As can be seen from Figure 10, with the increasing NS content, *K* was on the rise, especially when NS content was 2%. Results from Figure 11 and Table 4 illustrate that with the resting time developing, *K* of C–FA paste was on the decline, but *K* of C–FA paste incorporating NS was on the rise greatly.

#### 3.1.4. Rheological Index 

Shear thickening means that *η_0_* increases with shear rate developing, but shear thinning indicates that *η_0_* reduces along with increasing shear rate, representing different rheological properties of paste. In the H–B model, the rheological index (*n*) reflects the degree of paste shear thickening or shear thinning.

Figure 12 and Figure 13 show the influence of NS content and resting time on the rheological index (*n*). Results from Figure 12 indicate that with increasing NS content, *n* of paste decreased. Results from Figure 13 and Table 4 show that the *n* of C–FA paste was 1.689, 1.755, and 1.757 at 5 min, 60 min, and 120 min resting time, indicating that C–FA paste displayed shear thickening and with the increase of resting time the shear thickening was strengthened. However, after the incorporation of NS, the degrees of shear thickening of paste decreased. When NS content was 2%, *n* of paste was 0.947, 0.896, and 0.870 at 5 min, 60 min, and 120 min, which shows C–FA paste with 2% NS exhibited shear thinning and with the increase of resting time the shear thinning was obvious. 

### 3.2. Influence of NC on C–FA Paste Rheology

#### 3.2.1. Rheological Curve

Figure 14 displays the rheological curve of C–FA paste with NC at different resting times. It is clear from Figure 14 that with increasing shear rate, shear stress of all paste was on the rise. Moreover, the changes of rheological curves of C–FA paste with 0%, 1%, 2%, and 3% NC were tiny. Furthermore, with increasing resting time, the increasing rate of shear stress in the same shear rate gradually decreased. It is obvious the addition of NC did not change the rheological curve of C–FA paste significantly.

#### 3.2.2. Yield Stress

The influence of NC content and resting time on yield stress (*τ_0_*) are explained in Figure 15 and Figure 16. Table 5 displays the rheological parameters of C–FA paste with NC. As can be seen from Figure 15, when resting time was 5 min, along with increasing NC content, *τ_0_* of all paste was on a tiny decline, but when resting time was at 60 min and 120 min, *τ_0_* was on the rise slightly with increasing NC content. Figure 16 shows that with rising resting time, *τ_0_* was on the decline. In addition, the *τ_0_* of C–FA paste with 2% and 3% NC was higher than that of paste with 0% and 1% NC at 60 min and 120 min resting time. Table 5 also shows that the change of *τ_0_* was tiny when NC content was not more than 3%. 

#### 3.2.3. Plastic Viscosity 

The change of apparent viscosity (*η_0_*) on C–FA paste with NC at the different resting times are described in Figure 17. As can be seen from Figure 17, with increasing shear rate, *η_0_* decreased first and then increased and all pastes took on shear thickening. However, compared with C–FA paste, the addition of NC increased *η_0_* a little. When resting time was 120 min, it was hard to distinguish the influence of NC on *η_0_*. 

The influence of NC on plastic viscosity (*K*) is illustrated in Figure 18. Figure 18 shows that with increasing NC content, *K* of all paste was on a tiny rise. However, with increasing resting time, the *K* of all paste decreased, but the decrease was not obvious, as seen in Figure 19. 

#### 3.2.4. Rheological Index

Figure 20 and Figure 21 display the influence of NC on the rheological index (*n*) and the influence of resting time on *n*. Figure 20 shows that with increasing NC content, the *n* of all paste almost remained unchanged at the same resting time. Therefore, the addition of NC did not change the rheological properties of C–FA paste. Figure 21 illustrates that with the increase of resting time, the *n* of all paste slightly increased, indicating the incorporation of NC enhanced the degree of shear thickening of paste slightly.

### 3.3. Influence of NA on C–FA Paste Rheology

#### 3.3.1. Rheological Curve

Figure 22 displays rheological curves of C–FA paste with NA at different resting times. Figure 22 shows that with the increase of shear rate, rheological curves of C–FA paste incorporating different NA contents took on different shapes, suggesting the addition of NA into C–FA paste changed the rheological properties. First, the addition of NA increased shear stress of all paste. Second, the change of rheological curves of C–FA paste with 2% and 3% NA was obviously different from the rheological curve of paste with 0% and 1% NA. Especially, when NA content was 3%, the shear stress decreased first and then increased along with increasing shear rate, and the minimum shear stress was 20.3 s^−1^, 26.3 s^−1^, 33.9 s^−1^ at 5 min, 60 min, and 120 min resting time respectively.

#### 3.3.2. Yield Stress

Figure 23 and Figure 24 demonstrate the influence of NA content and resting time on yield stress (*τ_0_*). Table 6 shows the rheological parameters fitted from the H–B model. Figure 23 shows that with increasing NA content, *τ_0_* developed, and Figure 24 shows that with increasing resting time, the *τ_0_* of paste incorporating NA developed obviously. Table 6 shows that *τ_0_* of C–FA paste decreased along with rising resting time. However, when NA content was 1%, the *τ_0_* was 1.759 Pa, 2.447 Pa, and 4.976 Pa at the different resting times, and when NA content was 2%, the *τ_0_* as 44.5 Pa, 196 Pa, and 510 Pa at the different resting times. Extraordinarily, when NA content was 3%, the *τ_0_* was 753.8 Pa, 1021 Pa, and 1297 Pa at the different resting times, indicating *τ_0_* increased significantly after the addition of NA especially when NA content was beyond 2%. 

#### 3.3.3. Plastic Viscosity 

Figure 25a,b shows the change of apparent viscosity (*η_0_*) of C–FA paste with different NA and resting time. Figure 25 presents that there was a similar change of *η_0_* when NA content was 0% and 1%. Namely, with increasing shear rate, *η_0_* developed. However, when NA content was 2% and 3%, *η_0_* of paste decreased along with the increase of shear rate, showing that the rheological properties of the paste changed.

Figure 26 shows the influence of NA on plastic viscosity (*K*). Figure 26 shows that with increasing NA content, *K* increased, especially when NA content was 3%. Figure 27 shows that with rising resting time, *K* increased, and 3% NA content increased *K* significantly. Table 6 also shows that *K* of C–FA paste decreased gradually with rising resting time, but after incorporating NA, the *K* of paste increased greatly. 

#### 3.3.4. Rheological Index

Figure 28 and Figure 29 display the influence of NA content and resting time on the rheological index (*n*). As can be seen from Figure 28, with increasing NA content, the *n* of all paste decreased. When NA content was 2% and 3%, the *n* was not more than 1, indicating that C–FA paste with 2% and 3% NA were shear thinning pastes. Figure 29 illustrates that with the increase of resting time, the *n* of all paste slightly decreased except for C–FA paste. The *n* was always not more than 1 when NA content was 2% and 3%, showing that incorporating beyond 2% NA into C–FA paste changed the rheological properties of C–FA paste. 

## 4. Discussion

### 4.1. Rheological Properties of C–FA Paste 

The incorporation of FA can improve properties of cementitious materials obviously, including the fresh and hardened stages. The influence of FA on cementitious materials may be summarized as the morphological effect, micro-aggregate effect, and pozzolanic effect [39,40]. Among them, the morphological and micro-aggregate effects are mainly manifested by its lubrication and filling effect in cementitious materials caused by the characteristics of microsphere particles of FA. Some studies show that the particles shape and size distribution play important roles in the rheological properties of cement paste [19,41]. Paste mixed with a certain amount of FA will have better fluidity than pure cement paste. The addition of FA would decrease yield stress, plastic viscosity, and thixotropy of cement paste [10,41,42]. The apparent viscosity of C–FA paste takes on remarkable shear thinning behavior under very low shear rate, but changes to shear thickening behavior with the increase of shear rate, which shows C–FA paste is a shear thickening paste [10]. Also, in this study, along with rising resting time, yield stress and plastic viscosity of C–FA paste incorporating SP were on the decline, which may have been due to the retarding effect of SP [43]. As a result, the low plastic viscosity of C–FA paste would bring about segregation, bleeding, and coarse aggregate sinking in the fresh stage. Therefore, the incorporation of nanomaterials is a useful method to improve stability of C–FA paste.

### 4.2. Effect of Nanomaterials on C–FA Paste Rheology 

This paper focused on the influences of nanomaterials, including NS, NC, and NA on rheological properties of C–FA paste. From the results, nanomaterials led to different influences on rheological properties of C–FA paste. The influence mechanism of nanomaterials on rheological properties of C–FA paste is complex. Some reasons can be as follows.

#### 4.2.1. Specific Surface Area

The most important characteristic of nanomaterials is the huge specific surface area (SSA). Figure 30a,b presents the relationship between SSA and rheological parameters at 5 min resting time. In order to better compare the effects of different nanomaterials content and SSA on *τ_0_* and *K*, the ordinate is expressed by the natural logarithm of *τ_0_* (ln *τ_0_*). Figure 30 exhibits that nanomaterials with large SSA took on obvious effects on the improvement of *τ_0_* and *K*. The order of nanomaterials for the increase of *τ_0_* and *K* was NS, NA, and NC.

The main influence of nanomaterials on *τ_0_* and *K* of C–FA paste can be explained by the effect of a huge SSA and filling effect. The particle size of nanomaterials is very tiny, and these nanoparticles have large specific SSA and activities. In this study, the SSA of NS, NA, and NC were 150 m^2^·g^−1^, 120 m^2^·g^−1^, and 40 m^2^·g^−1^, respectively, which were much larger than those of C and FA. The SSA of NS and NA was larger than that of NC. Addition of these nanomaterials into C–FA paste not only changed the particle size distribution of suspensions but also enlarged the SSA of paste system. Therefore, the requirement of water to wrap nanoparticles in C–FA paste increased, leading to the reduction of free water in suspensions. However, this free water is helpful for increasing lubrication effect and decreases the friction force between particles, especially when these particles slip under shear rate [44,45], thus increasing the *τ_0_* and *K* value. The larger the specific surface area is, the more obvious the phenomenon is. These nanomaterials particles also can fill in the space between cement and fly ash, which makes the paste compact. Because the SSA of NC was much smaller than NS and NA, and the content of NC was not more than 3 wt%, the effects of SSA and filling of NC on C–FA paste were not obvious.

#### 4.2.2. Resting Time

With rising resting time, the rheological parameters of C–FA paste with different nanomaterials took on different changes. Figure 31 exhibits the changes of rheological parameters under the influence of resting time. In order to better compare the effects of resting time on *τ_0_* and *K*, the ordinate is expressed by ln *τ_0_* and ln *K*.

As can be seen from Figure 31, there were close linear correlation between ln *τ_0_* and ln *K* and resting time. Ln *τ_0_* and ln *K* of C–FA paste incorporating NS and NA developed with the rising resting time, but ln *τ_0_* and ln *K* of C–FA paste with the incorporation of NC decreased with the increase of resting time.

Resting time means the hydration time of cementitious materials. In the studies, in order to disperse these particles and maintain the fluidity of the system, some SP (0.4%, SP/C) was induced. In the early cement hydration process, some SP will be embedded in the hydration products and lose efficacy [46,47], resulting in the development of *τ_0_* and *K* with rising resting time. Lavergne et al. [38] and Long et al. [48] also found that introducing NS and NA into cement paste would accelerate hydration of the cement, leading to the formation of structure in an early stage and increase of *τ_0_* and *K*. The SSA of NC was smaller than that of NS and NA, and the content was low, therefore the effect of NC on rheological properties of C–FA was not obvious.

#### 4.2.3. Rheological Characteristic

Figure 32a,b shows the relationship between rheological index (*n*) and nanomaterials content at the different resting time. Figure 32a presents that C–FA paste was shear thickening paste, of which *n* was about 1.7. However, the *n* of C–FA paste decreased with the increase of NS and NA content. When NS and NA content were beyond 2 wt%, *n* was not more than 1, indicating paste taking on shear thinning. However, with increasing NC content, the *n* almost kept constant, meaning paste with NC still takes on shear thickening. Figure 32b exhibits that the *n* of paste incorporating 2 wt% NS, 2 wt% NA, and 3 wt% NA was lower than 1, which indicated these pastes took on shear thinning. Furthermore, with rising resting time, the *n* decreased, showing the shear thinning behavior was obvious. When resting time was 120 min, *n* of paste with 1 wt% NS was lower than 1, showing that with rising resting time, the rheological properties of paste with 1 wt% NS changed from shear thickening to shear thinning. However, *n* of paste with 1 wt%~3 wt% NC kept almost constant with increasing resting time, showing paste with NC was shear thickening paste. The incorporation of NS and NA changed the rheological characteristic of C–FA paste significantly. 

## 5. Conclusions

This study investigated and compared the influence of NS, NC, and NA on rheological properties of C–FA paste. Within the limits of this experimental study, the following conclusions can be drawn:

(1) With NS content and increasing resting time, yield stress and plastic viscosity of C–FA paste increase. When NS content is 2 wt%, the rheological index of C–FA paste is not more than 1, the paste takes on shear thinning, and the shear thinning becomes obvious with the increase of resting time. 

(2) The yield stress, plastic viscosity, and rheological index of the C–FA paste changes slightly with the increase of NC content in the range of 3 wt%. With the increase of the resting time, the yield stress and plastic viscosity of the paste decreases slightly, and the rheological index increases a little. The C–FA paste with NC presents shear thickening. However, the rheological curve of the C–FA paste with NC does not change significantly, and the rheological properties are not changed.

(3) With the increase of NA content and resting time, yield stress and plastic viscosity of C–FA paste increases, and rheological index decreases. When NA content is 1 wt%, the change of yield stress, plastic viscosity, and rheological index is not obvious. However, when NA content is 2 wt% and 3 wt%, yield stress and plastic viscosity increase significantly, and the rheological index drops below 1, showing rheological properties turning from shear thickening to shear thinning. Meanwhile, with the increase of the resting time, the shear thinning behavior of paste is obvious.

(4) Nanomaterials take different effects on the rheology of C–FA paste because of their larger SSA. The SSA of NS and NA is much larger than that of NC. The incorporation of NS and NA into the C–FA paste significantly increases the yield stress and plastic viscosity and decreases the rheological index. With the rising resting time, yield stress and plastic viscosity develop, but rheological index decrease gradually. The addition of NA and NS bring about the C–FA paste change from shear thickening to shear thinning, leading to the change of rheological characteristics. However, the rheological parameter of the C–FA paste incorporating NC does not change significantly. With the increase of resting time, the yield stress and plastic viscosity decreases slightly, the rheological index increases slightly, and rheological characteristics of paste take on shear thickening. The effect of large SSA on free water adsorption capacity, the effect of the acceleration of nucleation hydration of NA and NS on cementitious materials and the effect of adsorption of hydration products on superplasticizer are responsible for these differences.

## Figures and Tables

**Figure 1 materials-12-02598-f001:**
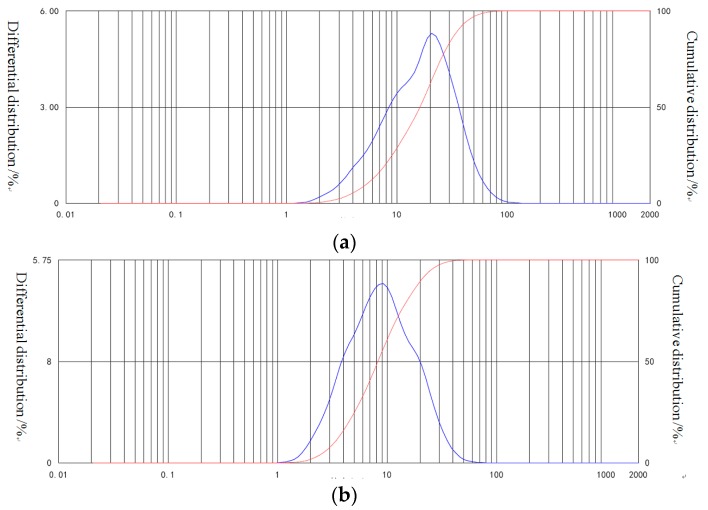
Particle sizes distribution of C and FA: (**a**) cement (μm)**;** (**b**) fly ash (μm).

**Figure 2 materials-12-02598-f002:**
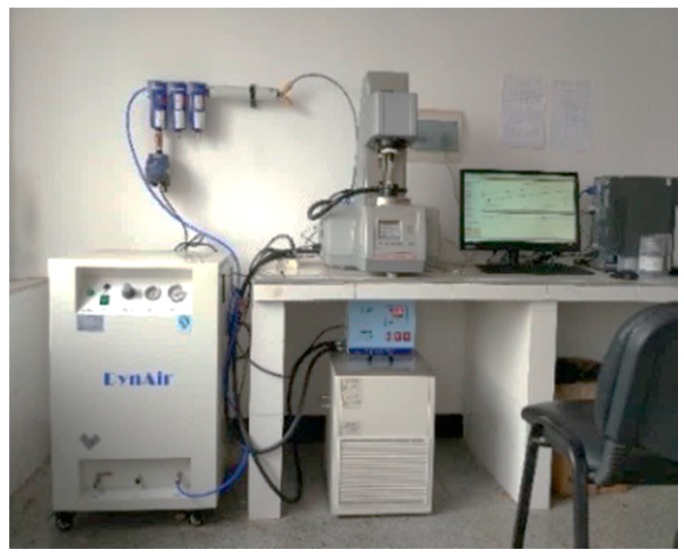
The employed rheometer.

**Figure 3 materials-12-02598-f003:**
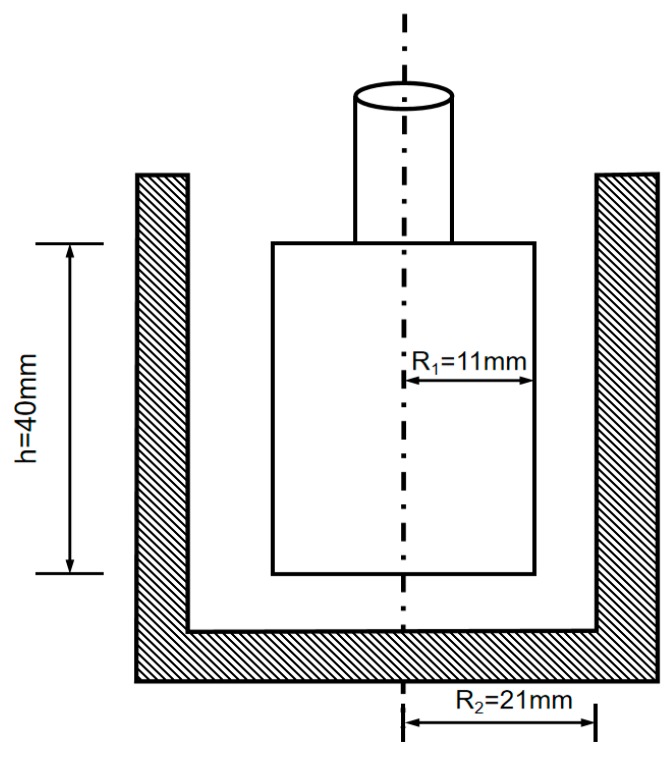
Coaxial cylinder size.

**Figure 4 materials-12-02598-f004:**
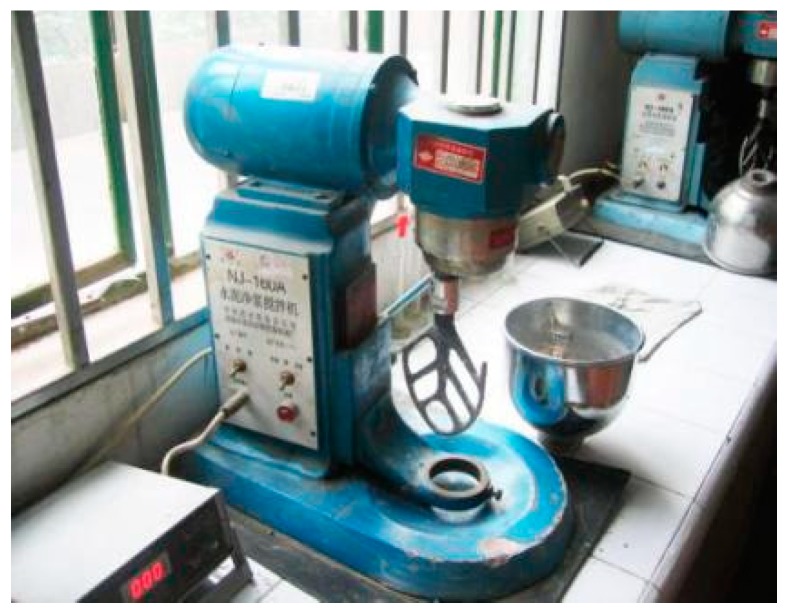
The electric mixer of cement paste.

**Figure 5 materials-12-02598-f005:**
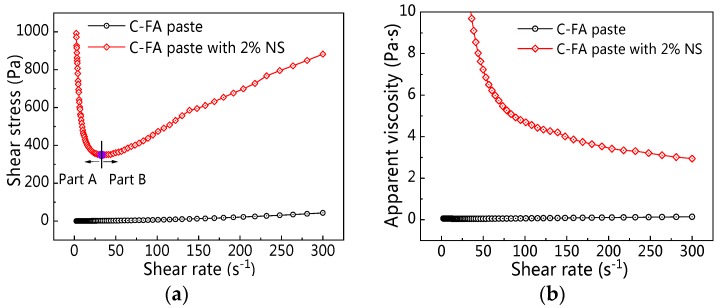
Rheological curves (at resting time 120 min): (**a**) shear stress vs. shear rate; (**b**) apparent viscosity vs. shear rate.

**Figure 6 materials-12-02598-f006:**
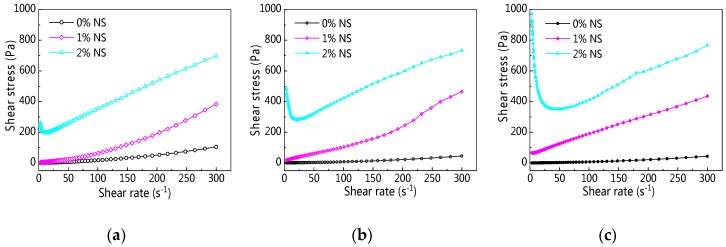
Rheological curve of cement–fly ash (C–FA) paste with nano-SiO_2_ (NS). (**a**) 5 min; (**b**) 60 min; (**c**) 120 min.

**Figure 7 materials-12-02598-f007:**
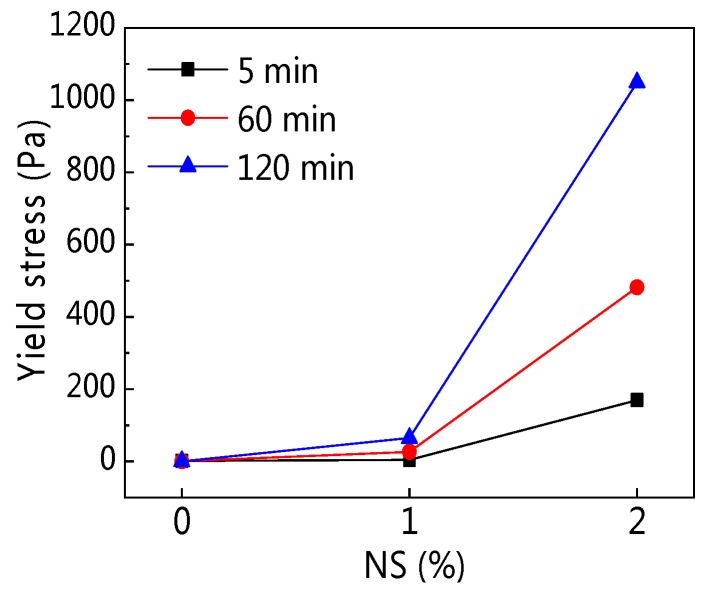
Influence of NS on yield stress (*τ_0_*).

**Figure 8 materials-12-02598-f008:**
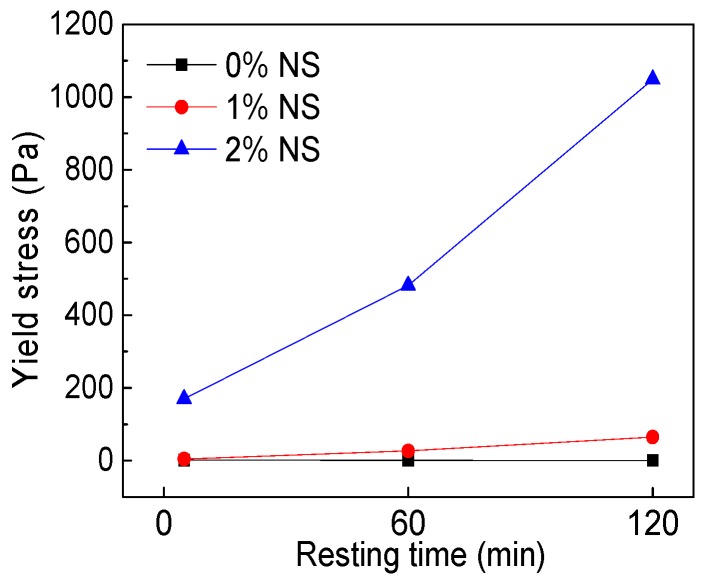
Influence of resting time on *τ_0_*.

**Figure 9 materials-12-02598-f009:**
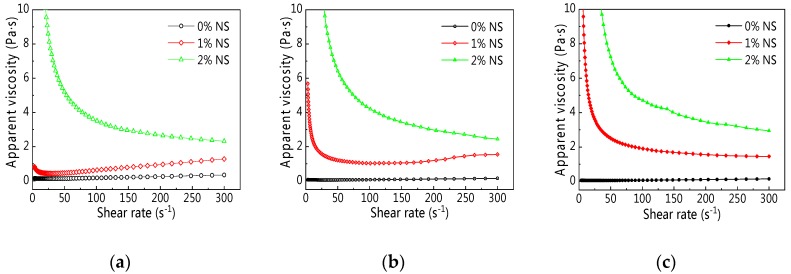
Apparent viscosity vs. shear rate under different NS content and resting time. (**a**) 5 min; (**b**) 60 min; (**c**) 120 min.

**Figure 10 materials-12-02598-f010:**
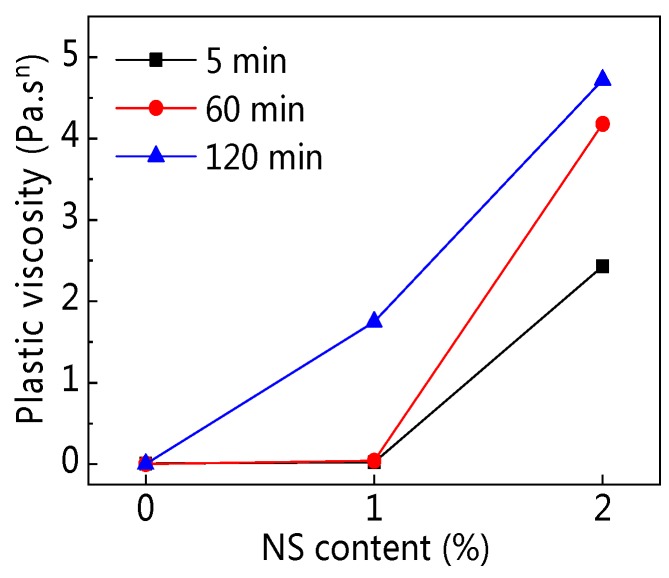
Influence of NS on plastic viscosity (*K*).

**Figure 11 materials-12-02598-f011:**
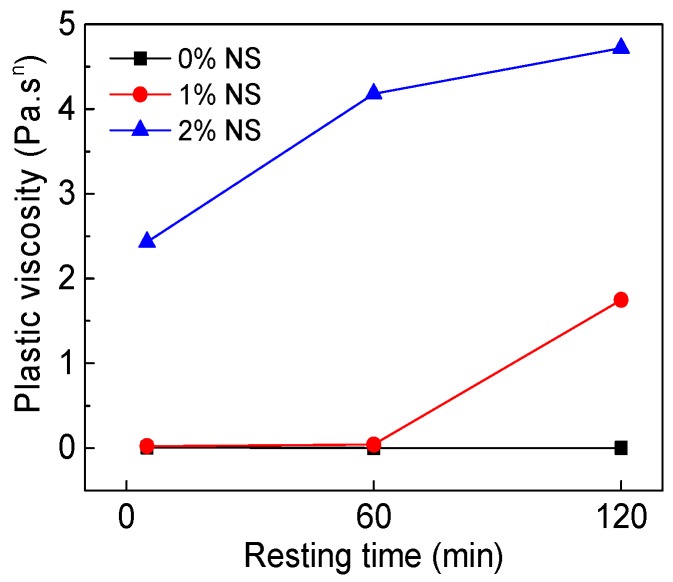
Influence of resting time on *K*.

**Figure 12 materials-12-02598-f012:**
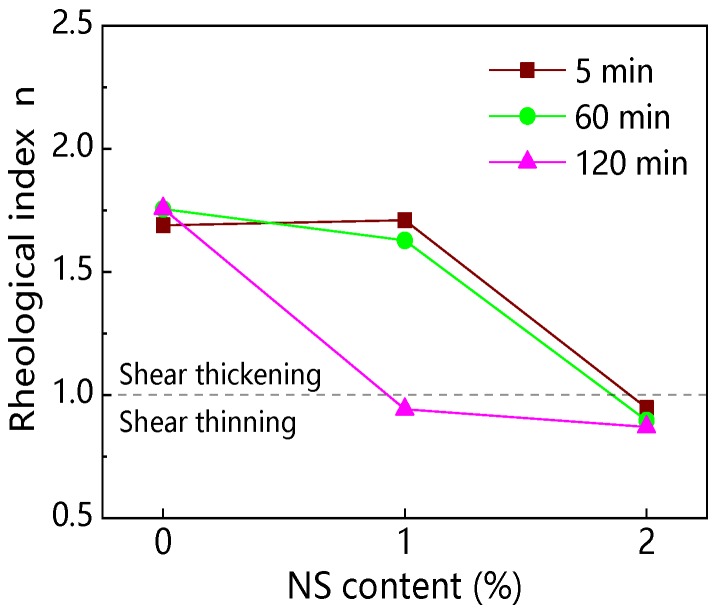
Influence of NS on rheological index (*n*).

**Figure 13 materials-12-02598-f013:**
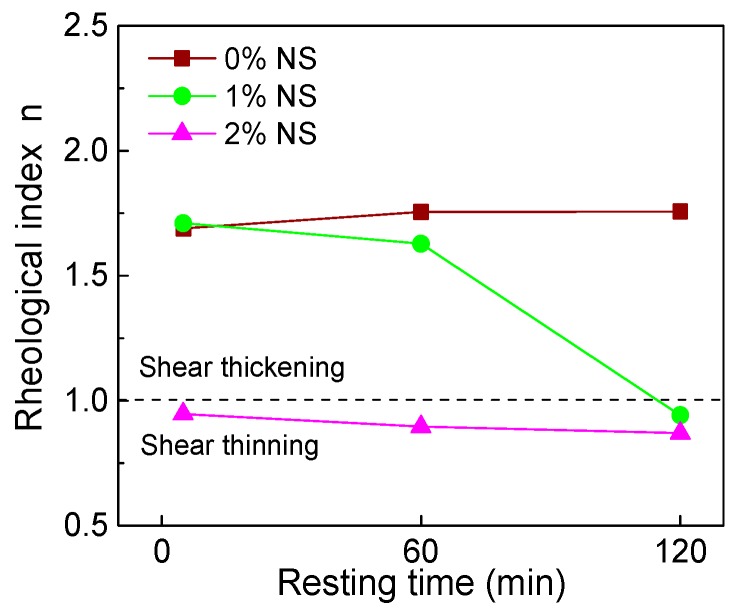
Influence of resting time on *n*.

**Figure 14 materials-12-02598-f014:**
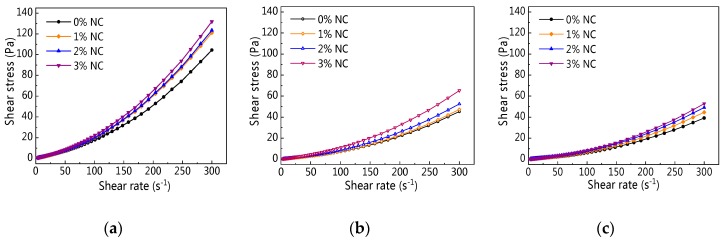
Rheological curve of C–FA paste with nano-CaCO_3_ (NC) under different resting time: (**a**) 5 min; (**b**) 60 min; (**c**) 120 min.

**Figure 15 materials-12-02598-f015:**
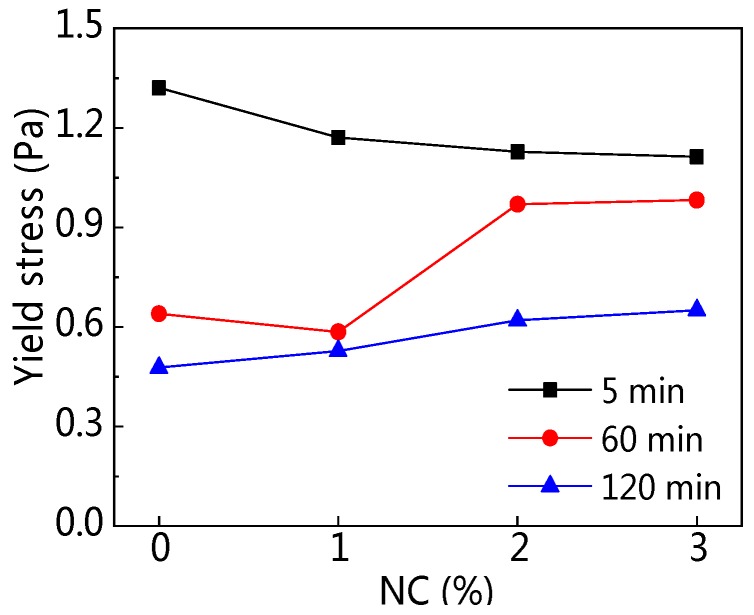
Influence of NC on *τ*_0_.

**Figure 16 materials-12-02598-f016:**
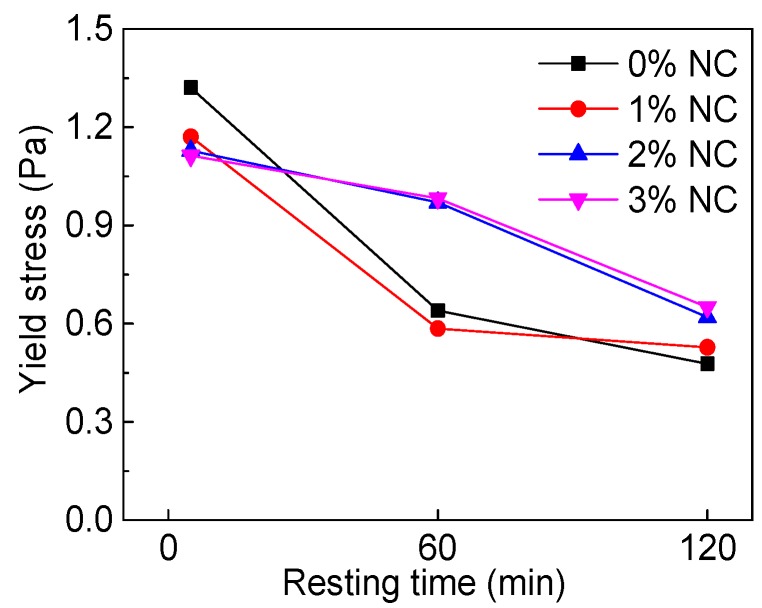
Influence of resting time on *τ*_0_.

**Figure 17 materials-12-02598-f017:**
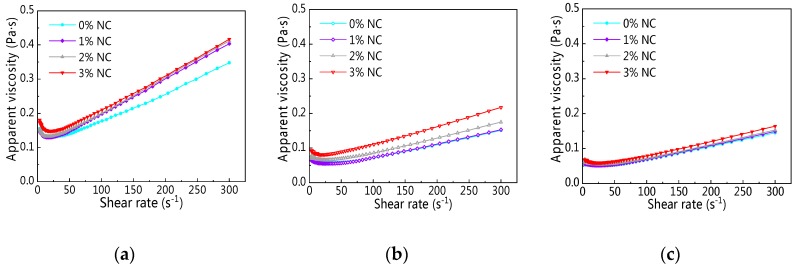
Shear rate vs. apparent viscosity under different NC content: (**a**) 5 min; (**b**) 60 min; (**c**) 120 min.

**Figure 18 materials-12-02598-f018:**
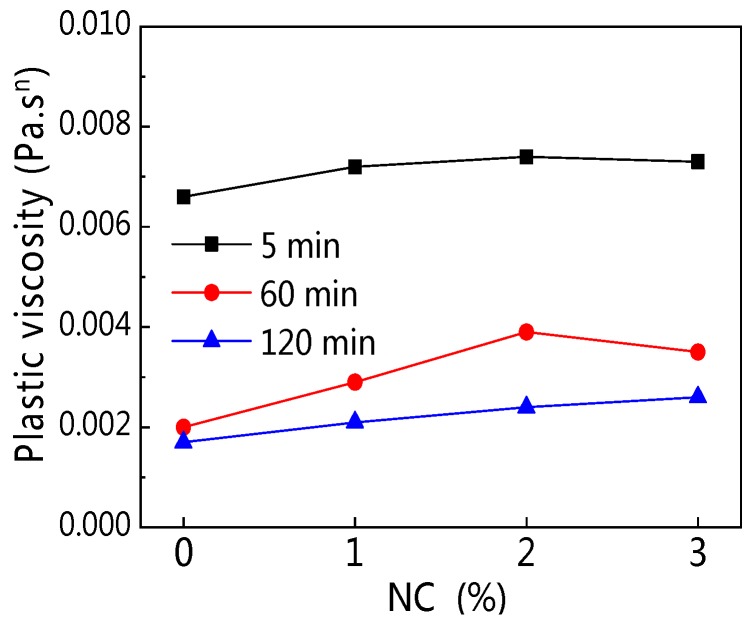
Influence of NC on *K*.

**Figure 19 materials-12-02598-f019:**
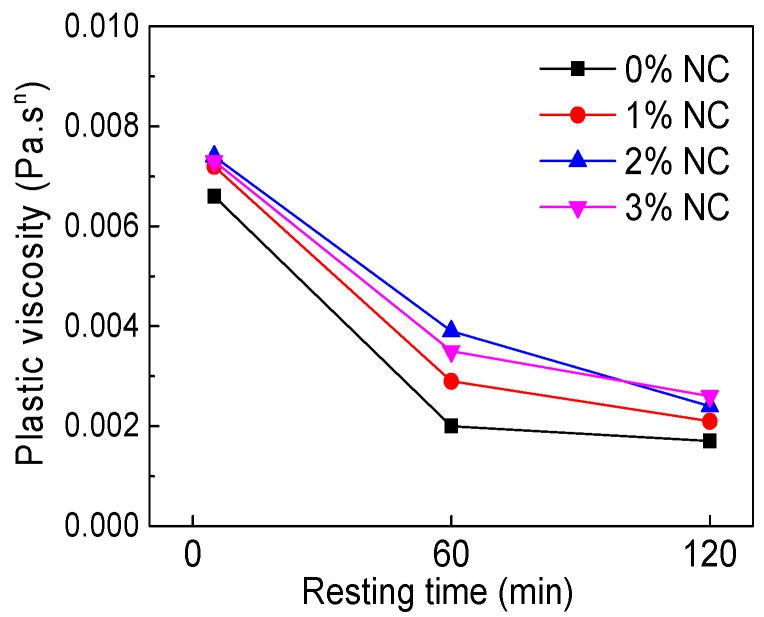
Influence of resting time on *K*.

**Figure 20 materials-12-02598-f020:**
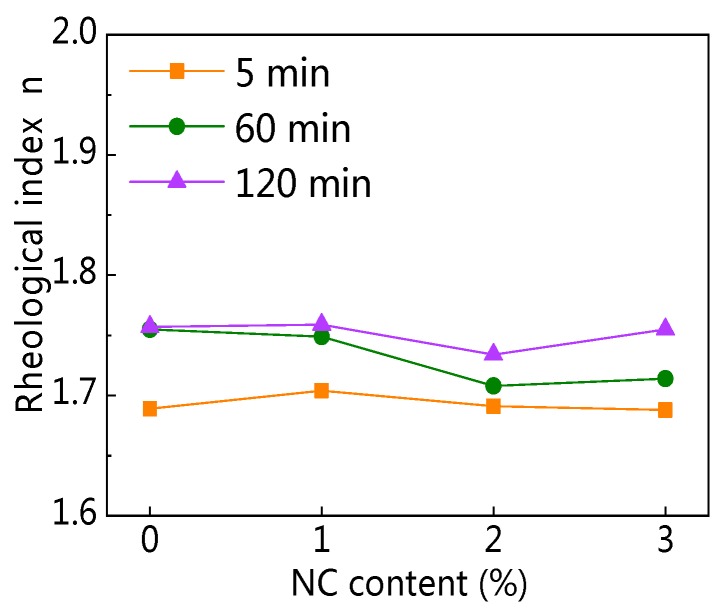
Influence of NC on *n*.

**Figure 21 materials-12-02598-f021:**
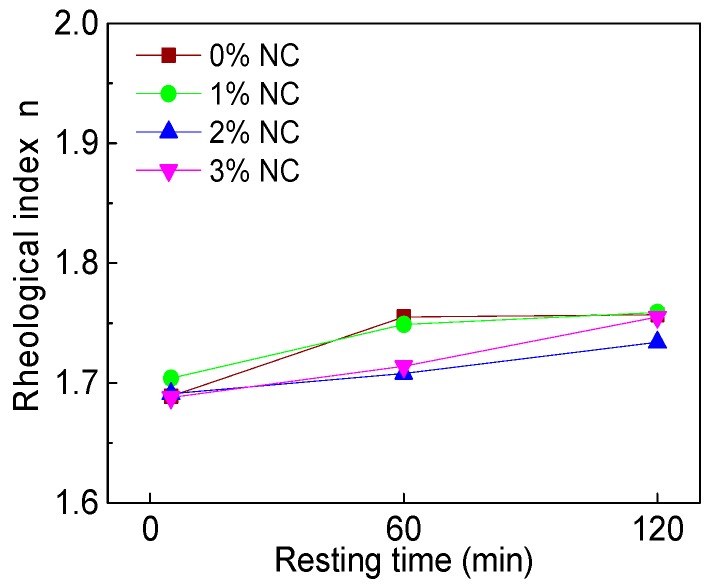
Influence of resting time on *n*.

**Figure 22 materials-12-02598-f022:**
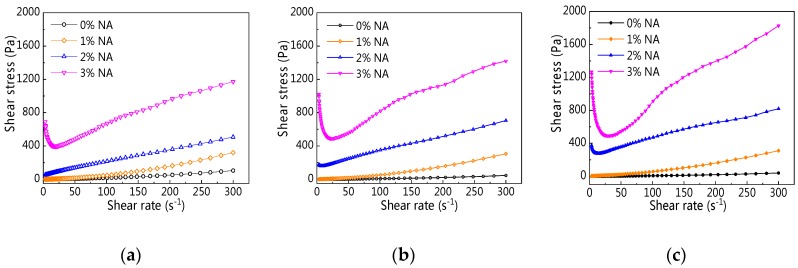
Rheological curve of C–FA paste with nano-Al_2_O_3_ (NA) at different resting times. (**a**) 5 min; (**b**) 60 min; (**c**) 120 min.

**Figure 23 materials-12-02598-f023:**
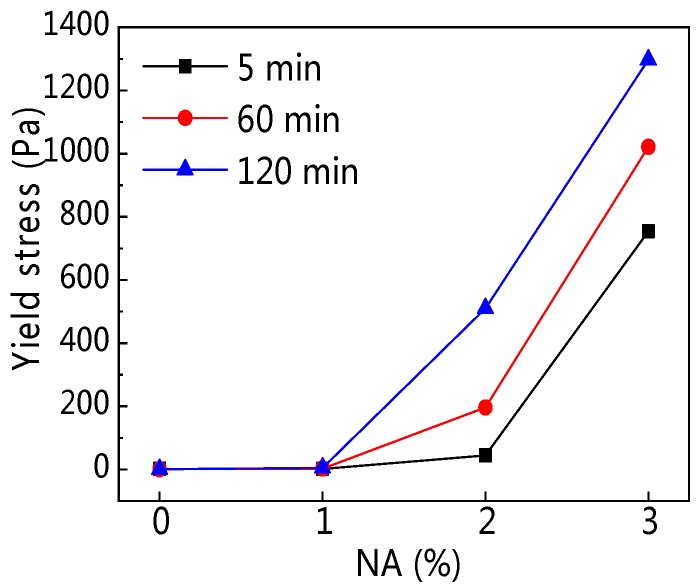
Influence of NA on *τ_0_*.

**Figure 24 materials-12-02598-f024:**
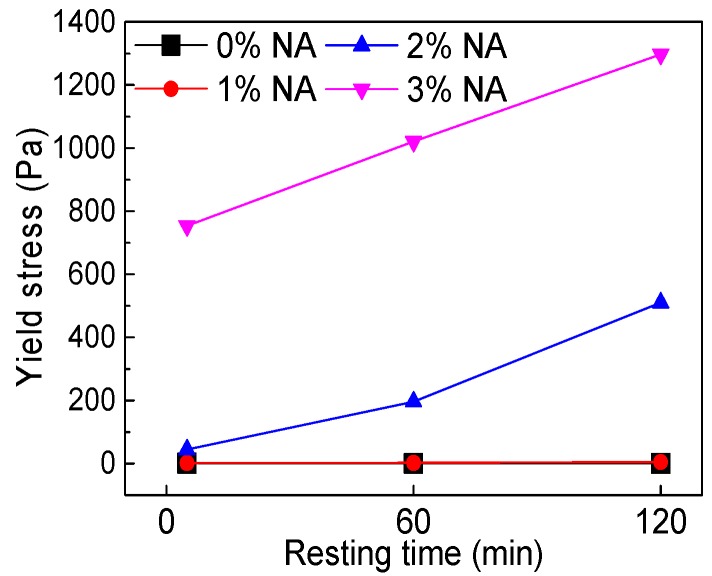
Influence of resting time on *τ_0_*.

**Figure 25 materials-12-02598-f025:**
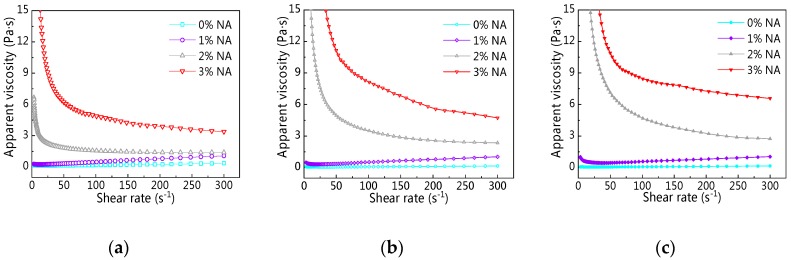
Apparent viscosity vs. shear rate under different NA content. (**a**) 5 min; (**b**) 60 min; (**c**) 120 min.

**Figure 26 materials-12-02598-f026:**
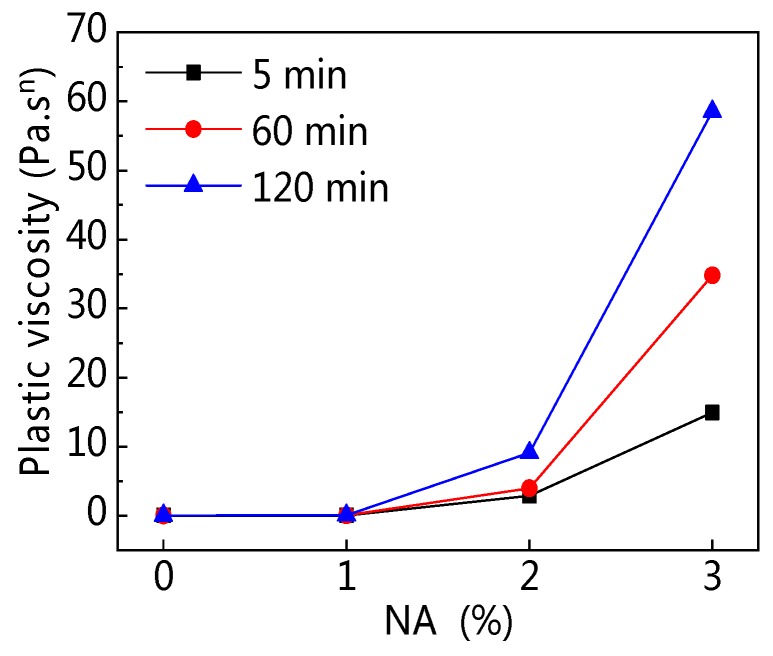
Influence of NA on *K*.

**Figure 27 materials-12-02598-f027:**
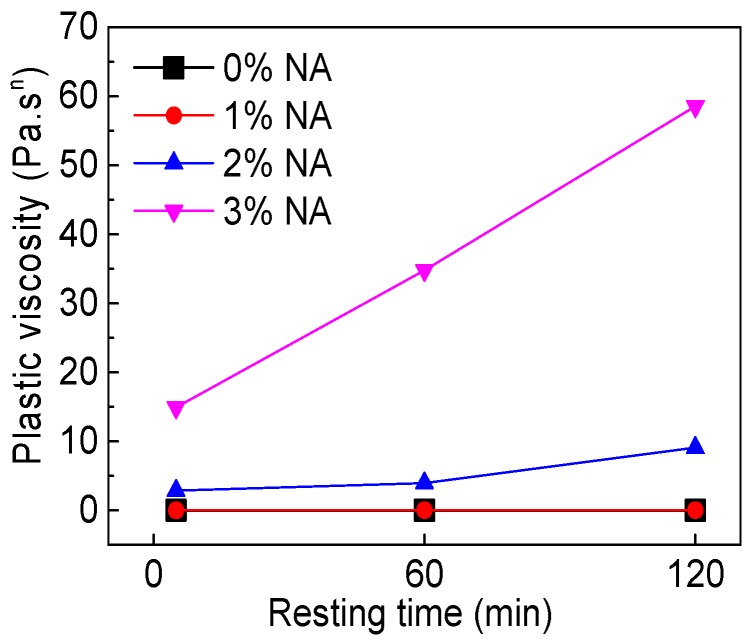
Influence of resting time on *K*.

**Figure 28 materials-12-02598-f028:**
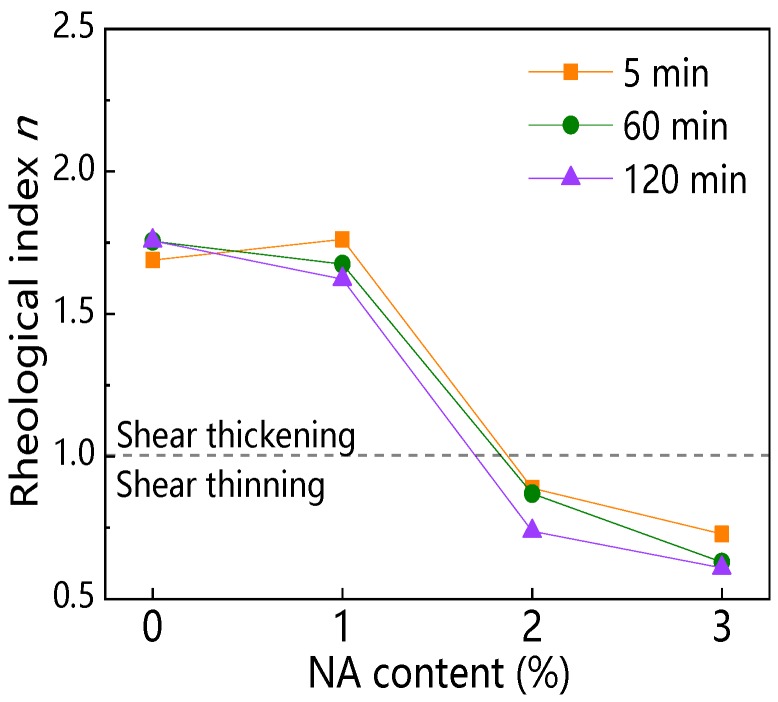
Influence of NA on *n*.

**Figure 29 materials-12-02598-f029:**
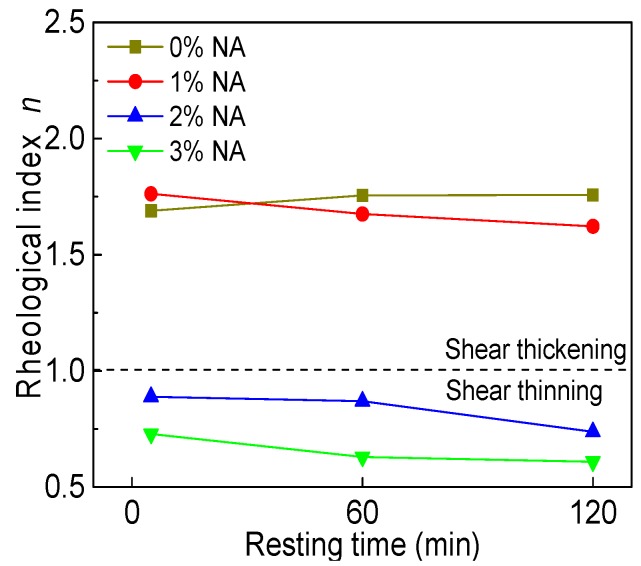
Influence of resting time on *n*.

**Figure 30 materials-12-02598-f030:**
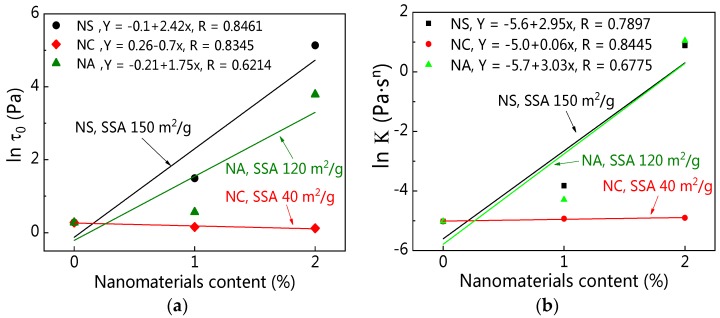
Specific surface area (SSA) and rheological parameters (resting time was 5 min.): (**a**) SSA and *τ_0_*; (**b**) SSA and *K*.

**Figure 31 materials-12-02598-f031:**
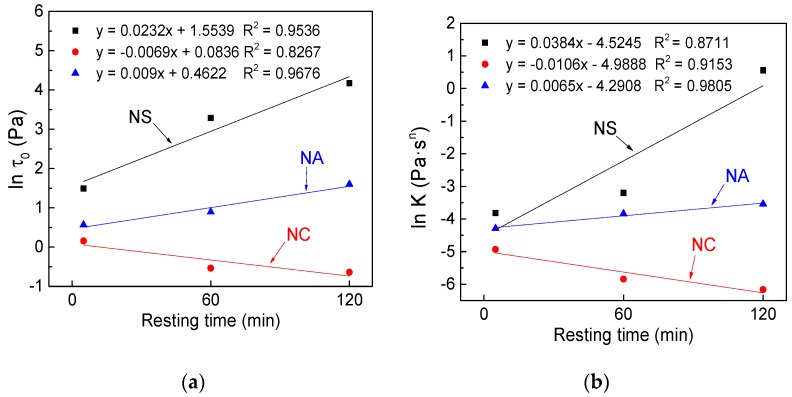
Influence of resting time on rheological parameters (1% content): (**a**) ln *τ_0_* and resting time; (**b**) ln *K* and resting time.

**Figure 32 materials-12-02598-f032:**
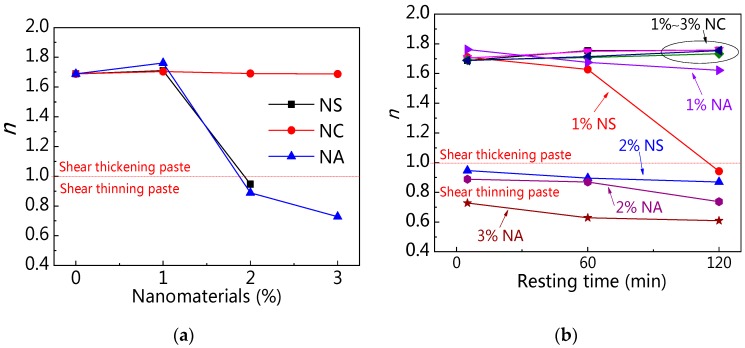
Influence of nanomaterials and resting time on rheological properties: (**a**) nanomaterials content vs. *n*; (**b**) resting time vs. *n*.

**Table 1 materials-12-02598-t001:** Chemical and physical properties of cement (C) and fly ash (FA).

No.	Cement (C)	Fly Ash (FA)
CaO	63.5	4.3
SiO_2_	20.6	52.3
Al_2_O_3_	4.6	26.1
Fe_2_O_3_	3.1	5.2
MgO	3.4	1.2
SO_3_	2.21	0.72
Loss on Ignition (%)	3.34	2.10
Alkali content (%)	0.59	1.19
Density (g/cm^3^)	3.05	2.38
Specific surface area (m^2^/kg)	339	463
28 d compressive strength (MPa)	50.3	-
C_3_A	7.4	-
C_3_S	56.7	-
C_2_S	15.1	-
C_4_AF	10.1	-

**Table 2 materials-12-02598-t002:** Physical properties of the nanomaterials.

	Crystal Type	Purity (%)	Average Diameter (nm)	Specific Surface Area (m^2^·g^−1^)
NS	/	99.5	30	150
NC	/	99.5	50	40
NA	α	99.9	20	120

**Table 3 materials-12-02598-t003:** Mix proportion of cement compound paste.

Sample	Mix Proportion (%)						
	Cement	FA	NS	NC	NA	Water	Superplasticizer
C–FA	75	25	0	0	0	32	0.4
C–FA–1%NS	74	25	1	0	0	32	0.4
C–FA–2%NS	73	25	2	0	0	32	0.4
C–FA–1%NC	74	25	0	1	0	32	0.4
C–FA–2%NC	73	25	0	2	0	32	0.4
C–FA–3%NC	72	25	0	3	0	32	0.4
C–FA–1%NA	74	25	0	0	1	32	0.4
C–FA–2%NA	73	25	0	0	2	32	0.4
C–FA–3%NA	72	25	0	0	3	32	0.4

**Table 4 materials-12-02598-t004:** Rheological parameters of C–FA paste with NS.

No.	Resting Time (min)	*τ_0_* (Pa)	*K* (Pa·s^n^)	*n*	R
**C–FA**	5	1.321	0.0066	1.689	0.9991
	60	0.6399	0.0020	1.755	0.9992
	120	0.4778	0.0017	1.757	0.9994
**C–FA–1%NS**	5	4.449	0.0219	1.710	0.9998
	60	26.72	0.0405	1.628	0.9935
	120	65.06	1.748	0.942	0.9995
**C–FA–2%NS**	5	170	2.43	0.947	0.9997
	60	482	4.18	0.896	0.9982
	120	1049	4.72	0.870	0.9955

**Table 5 materials-12-02598-t005:** Rheological parameters of C–FA paste with NC.

No.	Resting Time (min)	*τ_0_* (Pa)	*K* (Pa·s^n^)	*n*	R
**C–FA**	5	1.321	0.0066	1.689	0.9991
	60	0.6399	0.0020	1.755	0.9992
	120	0.4778	0.0017	1.757	0.9994
**C–FA–1%NC**	5	1.171	0.0072	1.704	0.9996
	60	0.5851	0.0029	1.749	0.9993
	120	0.5277	0.0021	1.759	0.9994
**C–FA–2%NC**	5	1.128	0.0074	1.691	0.9996
	60	0.97	0.0039	1.708	0.9992
	120	0.6201	0.0024	1.734	0.9993
**C–FA–3%NC**	5	1.113	0.0073	1.688	0.9994
	60	0.9827	0.0035	1.714	0.9992
	120	0.6506	0.0026	1.755	0.9993

**Table 6 materials-12-02598-t006:** Rheological parameters of C–FA paste with NA.

No.	Resting Time (min)	*τ_0_* (Pa)	*K* (Pa·s^n^)	*n*	R
**C–FA**	5	1.321	0.0066	1.689	0.9991
	60	0.6399	0.0020	1.755	0.9992
	120	0.4778	0.0017	1.757	0.9994
**C–FA–1%NA**	5	1.759	0.0137	1.762	0.9999
	60	2.447	0.0214	1.675	0.9998
	120	4.976	0.0291	1.622	0.9997
**C–FA–2%NA**	5	44.47	2.863	0.8885	0.9996
	60	196	3.949	0.8694	0.9989
	120	510	9.082	0.7374	0.9991
**C–FA–3%NA**	5	753.8	14.92	0.7281	0.9986
	60	1021	34.79	0.6291	0.9937
	120	1297	58.55	0.6090	0.9897

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
