# Peer review of "Influence of Nano-SiO2, Nano-CaCO3 and Nano-Al2O3 on Rheological Properties of Cement–Fly Ash Paste"

_materials, 2019, doi:10.3390/ma12162598_

Round 1

Reviewer 1 Report

The effect of nanoparticles on fly ash-cement paste was investigated. Incorporating the nanoparticlues such as nano-silica and nano-alumina increases the viscosity of paste. Regardless of the soundness and scientific justification of the manuscript, the findings provided in the manuscript are very trivial and already reported in various articles. Therefore, it is hard to recommend the manuscript to be published in the journal.

Reviewer 2 Report

The article presents research on the rheological properties of cement paste with fly ash (C-FA paste) containing diverse content of nano-SiO2 (NS), nano-CaCO3 (NC) and nano-Al2O3 (NA), and with the different resting time. Herschel-Bulkley model was applied to describe the relationships of shear stress vs. shear rate and apparent viscosity vs. shear rate. The authors emphasize that there are not many articles discussed rheological properties of cement pastes containing nanomaterials.

The article requires some refinements:

Line 38-39 -> the statement "The incorporation of FA leads to the decrease of yield stress, plastic viscosity and thixotropy …" should be supported by reference to literature.

Line 48 -> it would be valuable to reference also to publications where different type of SPs was used.

Line 90 -> there is lack of information how the physical properties of nanomaterials were obtained (whether they were tested by author or specified by the manufacturer)

Line 100 -> there is a lack of information about the measurement system and rheometer geometry, e.g., rotational viscometer with co-axial cylinders, size of the measuring gap

Line 105, 107, 108, 155, 224, 249, 250, 363 -> no space between the value and the unit

Figure 14, 16 -> no space between the value and the unit

Line 160 -> why there are different precisions in the presentation of the result of the yield value - it should be harmonize or justify.

Line 301 -> for the C-FA sample, the yield value decreases with time, but the differences are within the measurement uncertainty. In addition there is a lack of information how many samples were tested for each type of cement paste.

In my opinion, the article presents an interesting analysis desirable in the environment of researchers working on the use of rheological properties of cement pastes whit nanomaterials. Thus I recommend this work for publication after taking into account the above-mentioned comments.

Reviewer 3 Report

Please find attached a PDF file with my comments and suggestions for authors.

Round 2

Reviewer 1 Report

Please refine editorial errors in the manuscript.

Author Response

Thanks for the Reviewer’s suggestion. The editorial errors in the manuscript have been refined carefully.